# An Ecological and Neural Argument for Developing Pursuit-Based Cognitive Enrichment for Sea Lions in Human Care

**DOI:** 10.3390/ani14050797

**Published:** 2024-03-04

**Authors:** Peter F. Cook, Colleen Reichmuth

**Affiliations:** 1Social Sciences Division, New College of Florida, Sarasota, FL 34243, USA; 2Long Marine Laboratory, Institute for Marine Sciences, University of California Santa Cruz, Santa Cruz, CA 95064, USA

**Keywords:** sea lions, enrichment, cognitive enrichment, pursuit, hunting, goal-based cognition

## Abstract

**Simple Summary:**

In order to address the physical and mental needs of animals in captivity, scientists and animal care personnel have developed a number of enrichment strategies. These generally involve providing opportunities in the captive environment for the animal to engage in natural behaviors. Cognitive enrichment, that is, giving animals an opportunity to explore their environment and solve problems, is thought to ward off boredom and understimulation. The cognitive enrichment approaches used with captive sea lions and other marine mammals have tended to be similar to those developed for captive primates, even though the two groups of animals behave very differently in the wild. Here, we posit that the types of problems sea lions have to solve in the wild, for example chasing down and catching live prey, and the brains they have evolved to solve those problems, suggest a broader approach. We recommend the evaluation of enrichment built around target pursuit and flexible goal-based behavior in water.

**Abstract:**

While general enrichment strategies for captive animals attempt to elicit variable and species-typical behaviors, approaches to cognitive enrichment have been disappointingly one-size-fits-all. In this commentary, we address the potential benefit of tailoring cognitive enrichment to the “cognitive niche” of the species, with a particular focus on a reasonably well-studied marine carnivore, the sea lion. Sea lions likely share some cognitive evolutionary pressures with primates, including complex social behavior. Their foraging ecology, however, like that of many terrestrial carnivores, is based on the rapid and behaviorally flexible pursuit of avoidant prey. Unlike terrestrial carnivores, sea lions carry out this pursuit in a truly fluid three-dimensional field, computing and executing sensorimotor transformations from any solid angle to any other. The cognitive demands of flexible prey pursuit are unlikely to be fully elicited by typical stationary puzzle box style foraging enrichment devices or screen-based interactive games. With this species, we recommend exploring more water-based movement activities generally, and complex pursuit challenges specifically.

## 1. Introduction

Otariids, comprising 14 living species of fur seals and sea lions, are long-lived, gregarious, and highly trainable marine carnivores widely held in zoos and aquariums. Their complex social life, athletic and socially cooperative hunting behavior, and potentially wide home ranges in the wild present a strong contrast to the limitations of life in captivity [1,2,3]. Although sea lions are behaviorally flexible and generally adapt well to managed care [4], they may benefit from well-considered enrichment programs that provide meaningful opportunities to engage in species-typical behaviors. Given the complexity of their social and foraging ecology, as well as their apparent cognitive sophistication [5], they represent a potentially fruitful model for exploring new concepts and approaches to “cognitive enrichment” [6]. Here, we briefly address the goals of cognitive enrichment, discuss current approaches that have been used with sea lions, and suggest some ecologically valid alternative enrichment strategies.

Conceptions of animal welfare in captivity have typically focused on physical health, bodily integrity, and freedom from pain and fear [7]. Newer philosophical frameworks consider the opportunity for positive experience (flourishing) and agency [8,9,10]. In modern captive animal contexts, including zoos, aquaria, and rehabilitation and research institutions, animal welfare is usually promoted through providing access to healthy food and living conditions, veterinary care, and what is broadly termed “enrichment” While, traditionally, behavioral enrichment and other aspects of animal husbandry have been dealt with separately, there is clear potential for integration—almost any structured, complex interaction a captive animal has might be enriching. This includes participation in research, training for cooperative health and husbandry activities, and interaction with conspecifics. There is an argument for considering these factors holistically [11]. Most captive enrichment approaches presuppose that captive environments are impoverished—they do not afford the same range of opportunities for species-appropriate behavior as natural environments do. Enrichment interventions allow animals to express more “natural” and complex behaviors and have been shown by many measures to improve indices of animal welfare in captivity [12,13]. Importantly, but beyond the scope of this perspective, the philosophical value of centering “natural” or “wild” behaviors in enrichment programs has been challenged on a number of fronts [14]. These are productive arguments—here, we focus on enrichment approaches designed to engage an animal mentally.

While a subset of naturally occurring vertebrate behavior involves instinctual or “automatic” processes, a large portion of behaviors clearly involve flexible interaction with the environment and affordances therein. This flexible interaction is the specific domain of cognition, the set of mechanisms that allow animals to adapt in real time to unique situations. Psychologists and cognitive scientists define cognition as flexible information processing, and typically divide it into interrelated domains addressing information intake (sensation and perception), storage (memory), manipulation (working memory and decision making), and behavioral/motoric responses to those internal processes (action) [15,16,17]. Animals deploy all of these mechanisms when interacting with complex environments, and it is intuitive to ask what role the opportunity for cognitive deployment (colloquially known as “thinking”) plays in welfare. From a human perspective, consider boredom—a negative affective state characterized by insufficient cognitive challenge—and the relief and positive feelings associated with its alleviation by novelty and challenge. Whether other animals feel boredom or not is debated [18], but many animals do show an innate interest in novelty [19,20]. Further, neurobiological systems supporting affect and motivation are densely integrated with neurocognitive systems (e.g., frontostriatal contribution to learning and decision making [21])—clearly, novelty and familiarity, and thus information, have some affective value for mammals. Recently, scientists and animal care personnel have begun to question what role cognitive activity has in enrichment, and, in line with the goals of enrichment, what role it has in increasing animal welfare. Cognitive enrichment has been underutilized and understudied [22], though, importantly, many examples of enrichment activities and training interactions likely have under-considered cognitive ramifications (e.g., [23]).

Clark [24] defines cognitive enrichment broadly but sensibly as an intervention that gives animals opportunities to solve problems in their environment, resulting in some positive impact on welfare. As addressed above, typical enrichment is designed to allow “natural” behaviors or decrease stereotypical behaviors. Contrastingly, many examples of cognitive enrichment are designed to compel animals to deploy one or more of their general cognitive mechanisms (sensation, perception, memory, emotion, problem solving, decision making, and action), whether or not those mechanisms or the mode of deployment matches the animal’s species-typical cognition. Different species of animal produce different patterns of behavior, but do they differ in cognitive capability and deployment? That is, do they have different cognitive niches? The current formulation of cognition relies heavily on the modern history of automated computation, and thus tends to emphasize general-purpose mechanisms across use cases [25]. Some theorists have argued that animals, at least the vertebrates, do not differ substantially in their underlying cognitive capabilities [26]. This claim has been updated to suggest that mechanisms for cognition are largely conserved, even if capability differs to an extent between species [27]. However, it seems plausible that, just as variable body morphology is selected to suit an ecological niche, so too should cognitive capability be selected to suit a cognitive niche [28,29]. Different species encounter different problems in their respective environments, and different problems demand different solutions, potentially involving different cognitive processes. While evolutionary pressure on cognitive faculty may seem abstract, it is axiomatic in modern biology that cognition and behavior are completely dependent on physical substrates—most immediately the nervous system. Nervous systems are inarguably subject to evolutionary pressure [30], and while they show the conservation of certain patterns across species, such as separation into primary sensory fields and subcortical organization of afferent and efferent pathways, they also show notable differences in the relative size and complexity of organization [31]. In line with these principles, the leading theories explaining some of these brain adaptations privilege the types and difficulty of cognitive problems species encounter, for example, the relationship between social group complexity, foraging ecology, and brain size across primates [32].

If animals do indeed differ in cognitive mechanisms and capability, there is room to tailor cognitive enrichment to better allow each species to express a more species-relevant suite of cognitive mechanisms in more species-relevant contexts. To do so accords with the general goal of creating enrichment opportunities that allow the expression of natural behaviors. What then to make of the types of cognitive enrichment commonly provided to sea lions? There has been reasonable effort put into environmental enrichment [33,34,35,36]—that is, increasing the complexity of an animal’s captive habitat [12], often through use of an increased range of multimodal sensory cues. Although not typically framed in terms of cognitive enrichment, such environmental enhancement almost certainly has cognitive ramifications. In terms of providing direct and rewarded opportunities for interaction, however, the majority of documented enrichment approaches with sea lions, as with most captive species, rely on stationary objects and apparatus, and many of these involve small-scale motor manipulations [37,38]. These have tended to take the form of the “puzzle boxes” or “food toys” long used in studies of animal cognition [39] and enrichment [40]. There are many variants of puzzle boxes, but typically a closed container holds a food reward. In order to access this, the animal must enact a series of motor behaviors such as pulling, pushing, twisting, and shaking. While some carnivores have shown high capability for these tasks [41,42], one cannot help but note the extent to which these types of interactions seem to be tailored to the primate hand and the types of physical interactions it allows. On a conceptually similar note, there is an increased interest in screen-based interactive “games” for primate enrichment [43], and this approach has recently been used very effectively with sea lions as well [44]. Clearly, these types of interventions have potential benefits for any vertebrate, inviting novel problem-solving behavior. They may be optimally suited for “in between” times, when captive animals are not currently engaging in more physically active training. Notably, the Nvy sea lions featured in the above screen-based enrichment study spend significant time performing goal-pursuit tasks in the open ocean or during participation in research. Stationary games and puzzle enrichment devices are intuitive for humans to design and implement. However, it is incumbent on those housing marine carnivores such as sea lions to consider whether there are species-typical or relevant cognitive mechanisms that could be elicited by different types of environmental interactions. Here, we might start with first principles and begin by considering the ecological niche of the sea lion.

Sea lions are long-lived, socially complex species [45]. Compared to the phocids, they spend an extended time with their mothers before weaning, and rest and hunt in complex social aggregations. Sea lions hunt a wide range of fish and cephalopod species in the open ocean and in near-shore environments [1]. They may hunt in groups, and there is some evidence of cooperation during prey manipulation and capture [46]. In the open ocean, they can herd fish into bait balls and cut across them to maximize feeding efficiency. In near-shore environments, they have been observed to herd fish up against shallows and into alcoves before catching them. They are best characterized as flexible pursuit predators, and it is here that the misfit with stationary puzzle enrichment devices may be most clear. The evolutionary pressure for carnivores to catch other vertebrates has been proposed as a major driver of carnivore brain size and intelligence [47,48]. This represents a type of arms race, where prey evolve to get better at evasion, and their predators evolve to get better at countering that evasion. While there is clearly a raw physical component to this (speed, power, endurance), flexible hunters also rely on perception, motor learning, motor control, prediction, and rapid decision making, all clearly part of the cognitive domain [49]. Sea lions have large, chimpanzee-sized brains [50], and pinnipeds are one of the few clades in the “over four billion club”—animals with more than 4 billion neurons [51]. There is some tantalizing evidence that neural mechanisms for reward and motor control in pursuit carnivore brains have been differentially selected to privilege cognition for flexible goal-oriented behavior in comparison to fine-grained motor manipulation. In primates, including humans, the putamen, a portion of the striatum essential for reward and motor learning, is much larger than the adjacent caudate nucleus [52], which is essential for flexible goal-directed behavior and reward learning [53,54,55]. The pattern is reversed in coyotes and California sea lions, where the caudate dwarfs the putamen [56]. This gross relative morphometric difference strongly suggests differential selection for two different types of reward learning and related motor selection—small motor manipulation and tool use for primates, for example, which are used in solving puzzle boxes and to interact with touch screens, and flexible pursuit for the carnivores. This is not to suggest that primates cannot successfully pursue, nor that carnivores cannot successfully manipulate. Primates do sometimes chase and hunt prey, and, as noted above, carnivores are often successful with puzzle boxes. Rather, it suggests some degree of cognitive specialization, whereby neural resources have been allocated to the systems most relevant to each species’ cognitive niche.

Sea lions may have another neurobehavioral specialization for hunting. While most terrestrial pursuit carnivores chase prey—and thus have to make predictions and act to catch those prey—in a predominantly two-dimensional flat plane, sea lions catch prey in fluid three-dimensional environments where they are able to rapidly perform the sensorimotor transformations necessary to take their body from any solid angular orientation to any other (see [57]). There is some evidence that this type of sensorimotor transformation relies on the caudate nucleus [58,59]. Cook and Berns found intriguing preliminary evidence of elaborated caudate frontal connectivity pathways in California sea lions in relation to coyotes [56].

Considering the foraging ecology of the sea lion, one for which it may have evolved a specialized brain to support flexible target pursuit and sensorimotor transformation in water, it is striking how few published studies there are on enrichment with these species that involve related tasks. Hocking and colleagues [60] explored swimming and water-based enrichment approaches, including scattering fish in the sea lions’ pool and using sinking food toys, and found that individuals responded well to these approaches. However, while these do involve movement in the water during foraging, they do not involve complex flexible goal pursuit across three-dimensional space.

There is clear potential to design and assess novel enrichment approaches for captive sea lions that rely less on primate-centric small motor manipulations and more on flexible active pursuit. One of the most conceptually straightforward pursuit enrichment interventions is providing live fish prey to captive sea lions. Although there is a small amount of literature on the potential benefit of feeding live prey to terrestrial carnivores [61], and one paper on furnishing penguins with live prey [62], there seem to be no empirical assessments of the feasibility of doing so with pinnipeds for the purpose of enrichment (though see Dehnhardt and Schusterman for laboratory experiments involving live prey [63,64]). Live prey seek to evade capture, potentially engaging the pursuit predator’s cognitive apparatus for prediction and rapid decision making. They are also the most ecologically valid foraging intervention for captive sea lions.

There are potential risks and downsides to providing live prey, addressed recently by Marshal et al. [65]. There is increased possibility for injury when pursuing and consuming live prey, the potential of parasite transfer, and the public perception of the ethics of live-prey feeding must be considered at show facilities and zoos and aquaria. However, the potential benefits of this approach support the empirical assessment of feasibility and outcomes in a range of captive settings. Of course, provisioning with live prey is not the only method to provide flexible pursuit opportunities to captive sea lions. Just as terrestrial carnivores are sometimes provided with mechanized chase toys [66], a sea lion could be provided with a remote control or even automated water pursuit enrichment device. Remote control submarines have been successfully used in trained contexts to study wake following in sea lions as well as phocids [67,68]. Such approaches could be carefully adapted by, for example, operantly rewarding a sea lion for following a device. Safety, of course, would be paramount with such an approach, and the design and maintenance of appropriately durable underwater devices may be prohibitively expensive. A further intriguing possibility, and one very much in line with the value of cognition as a real-time adaptation mechanism, would be to provide a fully automated, AI-piloted vehicle, either with preset swim patterns or guided by machine learning in aimed at increasing evasion time. Such automated pursuit/avoidance agents have been explored virtually in video game development and in embodied agents by the military (e.g., [69,70]). A target that gets better at evading the better the sea lion gets at capturing it would certainly present a dynamic and species-relevant enrichment challenge. Importantly, this would allow the sea lion to develop and deploy novel hunting tactics, just as wild sea lions do. In addition to providing a chase enrichment to individual sea lions, some of these approaches could be used with groups of captive sea lions. In the wild, sea lions will sometimes hunt in groups, and doing so may have social emotional and social cognitive benefit. Simpler pursuit tasks are less logistically challenging to design and safety-proof. For example, dead fish can be propelled into a pool, as seen in [71], potentially providing an unexpected and engaging quick pursuit target. The goal should be to develop and assess different approaches for achieving feasibility, safety, and behavioral and health outcomes.

While flexible pursuit tasks may be most isomorphic with wild sea lion hunting behavior, and thus be most directly relevant to pursuit cognition, creative researchers and trainers can provide other goal-based swimming opportunities that privilege flexible and emergent cognition. One highly adaptable operant approach was first introduced to the public by Skinner [72]. By simply selecting and rewarding successive behaviors offered by the animal, an observant trainer can shape subsequent behavior toward a wide range of divergent outcomes. We used this approach when working with captive pinnipeds at the Long Marine Lab in enrichment, husbandry, and research training contexts. We sometimes refer to this approach as the “training game” and it proceeds as follows. The trainer typically pre-selects a goal for a session. They then position themselves near the animal enclosure with a food reinforcer and proceed to ignore the animal. For animals with experience at the game, the trainer’s sitting in a particular location or wearing of a particular hat can serve as discriminant cues that the game has begun. But even for naive animals, if the trainer is patient, the animal will eventually stop orienting toward them and go about their own business. At this point the game begins. The trainer simply selects whatever behavior the animal offers that seems like it might potentially lead toward the final goal, and rewards the animal with a fish. If the animal then begins to orient to the trainer, one simply waits until they again redirect their attention, and then the trainer again looks for opportunities to reward the animal. Most animals begin to understand the game within a few such well-timed rewards, at least to the point of scanning the environment for interaction opportunities that may lead to reward, as opposed to passively sitting and waiting for a signal from the trainer. In this operant, environmentally focused mindset, individuals can offer behavior or creatively explore. They can of course also attempt to “guess” the correct answer based on prior experience. The trainer has some capability with reinforcement timing in pushing the animal toward or away from any of these strategies.

A typical game might go as follows. An experienced trainer approaches the sea lion’s enclosure with the goal of getting the sea lion to float upside down in the center of the pool with its rear flippers extended above the water. As the sea lion moves away from the trainer, the trainer drops a fish into the water. After a few such reinforcements, the trainer becomes more selective—now, the sea lion will only receive a fish for moving into the water. Relatively quickly, the sea lion may begin swimming in a restricted space closest to where they were first reinforced. If they duck their head, the trainer perhaps reinforces that with the aim of shaping an underwater float and then progressively relocating it to the center of the pool. If the animal instead moves closer to the center of the pool, the trainer can select that with the aim of shaping the location first and then working on the underwater float. By successive approximation (rather like a game of hot and cold, where a fish represents hot and no response means cold), the animal can be gradually moved toward the chosen behavior. However, the animal must do the work of sampling the environment to find the right behavior. This promotes behavioral flexibility and open engagement with the environment. We have used this approach to successfully train sea lions to perform a repeated porpoise behavior, to jump through a hoop, and to fetch items from the bottom of a pool. We have also used the same approach with wild sea lions in rehabilitation to train participation rapidly and remotely in a T-maze memory assessment task [73].

Note that, from the perspective of enrichment, a trainer may enter the enclosure with no specific goal except to increase the variability of behavior in the animal, or to reward whatever behavior the animal seems most interested in performing, and then let the animal truly explore the environment with agency nd some measure of self-determination. An important caveat should be noted as follows: animals may become frustrated when they cannot discern the goal of the training session, when sessions are too long or occur too frequently, or when the trainer does not provide the density of reinforcement to which the animal is accustomed. Of course, experiencing and potentially overcoming frustration can, in itself, be cognitively rewarding [74]. Trainers might begin with simpler goals, and, as the animal learns the “rules” of the game, attempt more complex sessions over time. These training game sessions might proceed similarly to “do something novel” tasks used with dolphins and walruses [75,76]. Regardless of approach, this type of exploratory training interaction can allow complex, flexible, and emergent goal-based behavior in the water, which likely taps into some of the same core cognitive mechanisms that make sea lions effective pursuit hunters.

While flexible goal pursuit in the water column seems, on its face, to match the cognitive ecology or niche of sea lions, this is no guarantee that related captive activities will improve animal welfare. Animals may engage in many tasks if their daily diet depends on their doing so, but this tells us little about their wellbeing. Cognitive interaction may, intuitively, be positive or negative. For example, imagine a human child’s feeling on playing a complicated but fun game vs. taking a difficult mathematics test—both tasks are cognitively challenging but with opposite emotional valence. The long-term health benefits of specific enrichment approaches with animals are hard to measure, always being multi-factorial. We do have some evidence that sea lions tend to be remarkably persistent when faced with difficult cognitive challenges [5], and the innovative computer-based enrichment study discussed above [44] found that cognitive challenge was no deterrent to sea lions engaging with a device. There is evidence in bears, another canid carnivore, of “contrafreeloading” behavior [77], that is, choosing to engage in operant behavior to obtain food when other food is freely available [78], and we have seen similar behavior in captive sea lions. We should not assume that sea lions will shy away from difficult tasks. One simple and underused approach to assessing enrichment benefit is to let an animal choose which task it will engage in. The expression of agency by a captive animal can be instructive (and potentially rewarding and enriching in its own right [79]) and pulls us away from our anthropomorphic tendency to privilege primate-style tasks that seem interesting to us. Of course, one possibility is that the novelty of such tasks makes them appealing to some sea lions, but choice paradigms could provide evidence either way. Alternative enrichment opportunities with similar rates of reinforcement can be presented in set locations. Following some experience of both separately, animals could be given a forced choice between them. Such balanced choice-based assessments could help us to find and tune approaches most conducive to sea lion engagement.

Aside from immediate cognitive engagement, there are other potential benefits to active pursuit and goal-based swimming enrichment for sea lions (as well, perhaps, for other pinnipeds and dolphins). Increased swimming provides exercise opportunities to a species that, in the wild, is extremely active. The benefits of exercise for captive animals are manifold. Further, increased movement has been shown to promote neurotrophic factors and even neurogenesis in mammals [80,81]. This may help slow neurobiological damage and decline [82], which may be particularly meaningful for long-lived animals. Of note, many captive sea lions come to captivity from the wild, where they may have been exposed to environmental neurotoxins contributing to medial temporal lobe epilepsy [83]. These brain areas are also those shown to be most responsive to the health-promoting effects of exercise. Experimental rodent data indicate that exercise drives increased neuroplasticity in animals with epilepsy [84]. A recent longitudinal imaging study of a long-term captive fur seal with this toxic exposure demonstrated that significant brain damage accrued over his lifetime in the medial temporal lobe. The damage progressed, even with consistent veterinary care and supportive therapy [85]. For captive sea lions, including those with medial temporal insult, variable exercise may have targeted therapeutic relevance.

## 2. Conclusions

Sea lions are commonly selected for captivity, both because of their high trainability and charisma and also their susceptibility to human-driven environmental change. We owe it to them to continue seeking out and empirically assessing the best possible enrichment approaches while optimizing welfare and minimizing stress. There has been a deficit of documented interventions that provide sea lions with opportunities to participate in cognitively engaging activities that mirror or substitute for what they might experience in the wild. Given the degree to which their foraging and social ecology center around hunting, and their apparent neurobiological adaptations for flexible pursuit behavior, sea lions may respond particularly well to enrichment tailored to promoting dynamic goal-driven behavior in the water. Further exploration of this topic is needed.

## Data Availability

Data are contained within the article.

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
