# Peer review of "An Ecological and Neural Argument for Developing Pursuit-Based Cognitive Enrichment for Sea Lions in Human Care"

_animals, 2024, doi:10.3390/ani14050797_

Round 1
Reviewer 1 Report
Comments and Suggestions for Authors
This is an interesting an important topic in zoo animal welfare. I agree that particularly with the species discussed, research on enrichment and promoting positive welfare outside of training sessions is lacking, so find this commentary a welcome addition to the field. The commentary provides a solid foundation of cognitive enrichment and its’ role in animal welfare and the theoretical framework underpinning it. There are some minor grammatical and use of English errors for example, lines 35 and 64 and therefore, I recommend that the manuscript is proof-read in its entirety to address these minor issues. The final part of the commentary discussing meeting the needs of animals brought from the wild could be clarified further, if the focus is on animals which will not be re-released then this should be stated and reaffirmed that the commentary focuses on animals permanently under human care. Alternatively, if this section is focusing on animals to be released, I feel this should be removed. Overall, I feel that this commentary will be thought-provoking to many in the industry from both research and practical perspectives, with a reach beyond pinnipeds. Due to this, I recommend that this commentary is accepted for publication with minor revisions.
Comments on the Quality of English Language
Overall the language used is appropriate, but the manuscript would benefit from proof-reading for minor grammatical errors and in some areas, plainer English could be used to improve readability.
Author Response
I appreciate the reviewer's feedback. I have proofread the document again and found a few small errors. Of note, line 35 had no error -- although the AI grammar checker in word processing software believes "well considered" should be hyphenated, "well" is an adverb, and proper American English does not hyphenate adverbial phrases. In addition, the grammar on line 64 was also correct, although the sentence structure was complex, so I have broken this into two shorter sentences.
I have clarified that the second-to-last paragraph is referring to long-term captives, not short-term release candidates.
Reviewer 2 Report
Comments and Suggestions for Authors
Overall, outstanding contribution, flawlessly written (I don’t have a single edit), clearly explained, and well-reasoned. A few comments for the author, that I would offer over a beer, but none that require any modification to the manuscript. Publishable just as it is.
I’m sure the references cited do emphasize the elicitation of “natural” behaviors as somehow fundamental to enrichment. I don’t know if I agree. There are natural behaviors that become maladaptive in a zoological setting, e.g., oral exploration that becomes cribbing, con-specific aggression that is amplified in the relative confines of an enclosure. These natural behaviors don’t enhance welfare. I would argue that complexity of behavior is equally valued for enrichment. I don’t think the author would disagree, but “natural” was discussed a lot more than “complex.”
I appreciated the detailed description of the training game and yielding control. It will serve as a good resource for trainers/keepers unfamiliar with these techniques, improving the quality of their interaction with their charges. And I agree that hunting for solutions like these probably engages many of the same cognitive muscles as hunting for elusive prey. However, they require the participation of a trainer, and that is a limited resource for most holders of marine mammals. A perhaps unspoken rationale for many enrichment activities is enhancing the time between training sessions.
Navy sea lions spend a good portion of their day hunting targets in the open sea, then return to their home enclosures, where live fish are abundant and largely ignored. They also sleep more than they’re awake. Still, we wanted to give them more opportunity for engagement in the enclosures. That’s why we called our computer interface the Enclosure Video Enrichment system. We hope it will be of value for sea lions in zoos and marine parks that don’t get to go to sea, and don’t have access to live fish. We wouldn’t require an IACUC for live fish procured just to be placed into an aquarium, or even released into a marine mammal pool. We have done IACUC protocols for fish when we received funding to do experiments on the fish themselves.
Working with underwater drones is challenging and expensive. The ones Navy sea lions interdict at sea for practice cost $175K each and requires a full-time technician to repair the damage the sea lions do to them. It’s possible some hobby shop drone could survive a pool, but I don’t know how elusive it would be, or how much contact with the sea lions it would survive. But this suggestion made me remember using a big, monkey-fist knot of rope on a long line and trying to pull it through the water faster than Lou Herman’s dolphins could chase it down. The dolphins usually won. And at Epcot, we had a game where a person on deck would chase and try to tag the dolphins with a long pole with a padded end. The dolphins could easily avoid being tagged but would come close enough to make being tagged possible, then glide just out of reach. And then, through good mammalian body language, the game would switch, and they would try to catch the end of the pole, while the person tried to keep it away from them. The game was so simple and safe that even a naïve volunteer could do it (i.e., not resource limited). But after a visitor observed us through the underwater windows and complained that we were “poking the dolphins”, the bureaucracy shut that game down.
Author Response
love these comments, and greatly appreciate the creative/constructive way they're framed! (and of course would be happy to discuss over a beer at whatever future point that might be convenient for both of us).
I've made a few welcome additions to the manuscript in response to some of these points.
I agree 100% that "natural" is not synonymous w/ "beneficial" or "healthy." I've included a reference and clarification of this point at the beginning of the introduction.
I've also added some further language about the value of the navy computer interface protocol for "in between times," and mentioned in general how different enrichment strategies may be necessary/optimal for different housing situations and circumstances.
I've further addressed some of the potential logistical hurdles for using underwater chase devices, and appreciate the applied perspective on this.
Again, many thanks to the reviewer.
Reviewer 3 Report
Comments and Suggestions for Authors
An interesting commentary linked to the literature discussing approaches to captive animal enrichment. The arguments for a more considered approach to the cognitive abilities of pinnipeds when designing enrichment is a useful addition to this field.
There are a few topics that might be considered to add strength to this commentary. Particularly to elaborate on the section Ln 43 – 53.
1. Consider integrating the concept of behavioural-based husbandry (see Bacon, H. (2018). Behaviour-based husbandry—A holistic approach to the management of abnormal repetitive behaviors. Animals, 8(7), 103.)
2. Further acknowledgement of the challenges, in terms of welfare outcomes, of focusing solely on behaviours adapted to thrive in a wild setting (see Learmonth, M. J. (2019). Dilemmas for natural living concepts of zoo animal welfare. Animals, 9(6), 318.) From a welfare perspective, it would be beneficial to reference that the captive environment comes with different cognitive challenges, and perhaps address how cognitive enrichment might be used to combat some of these challenges, in addition to facilitating a wild type behavioural (and cognitive) profile.
3. As a number of approaches to cognitive enrichment are proposed, there need to be further consideration of enclosure design, group structure and safety aspects of the proposed mechanical and training solutions. The these suggestions are afforded lengthy descriptions that less experienced individuals may wish to try, so relevant cautions around possible risks to pinnipeds and humans using these techniques ought to be considered.
Structure:
It would be of benefit to consider the concept of enrichment a little more fully in the introduction, e.g. by mentioning the physical benefits that enrichment may offer, alongside behavioural and cognitive outcomes. These are briefly referred to towards the end but by addressing early on might demonstrate a more complete overview of the topic of enrichment for welfare, prior to delving further into the shortcomings of enrichment in pinnipeds.
Other points:
Ln 120: full stop missing after references mid-way through this line.
Ln 246 – 272: This section might benefit from some additional reference to learning theories, motivation and frustration, in relation to the proposed training 'game', to add strength (and a more critical appraisal) to the argument of such a game for welfare reasons.
Ln 309: typo 'progressed'?
Ln 309: This source (76) appears to be used to justify that exercise and cognitive enrichment would have helped the seal in question, but that is not clear from the results of this study. Consider more conservative wording here.
Author Response
I thank the reviewer for these constructive comments.
I've included the suggested reference on behavioral husbandry and briefly addressed. It does add meaningful breadth to the beginning of the paper.
I've also put a bit more in the beginning about the potential exercise benefits of enrichment.
I also agree w/ the reviewer that I could have done more to address plausibility and safety of some of the suggested approaches, and I have done so in the revision.
I have added a few points on the potential for frustration and confusion in relation to the training game, as suggested.
"progressed" here is used in the sense that the damage worsened over his lifespan. I agree that slightly more clarification was needed here to address the potential value of exercise enrichment for animals w/ MTL insults, and have provided it.